# Effect of an Immunomodulatory Feed Additive in Mitigating the Stress Responses in Lactating Dairy Cows to a High Concentrate Diet Challenge

**DOI:** 10.3390/ani12162129

**Published:** 2022-08-19

**Authors:** Damiano Cavallini, Ludovica M. E. Mammi, Alberto Palmonari, Ruben García-González, James D. Chapman, Dereck J. McLean, Andrea Formigoni

**Affiliations:** 1DIMEVET—Dipartimento di Scienze Mediche Veterinarie, Università di Bologna, 40064 Bologna, Italy; 2Phibro Animal Health Corporation, Teaneck, NJ 07666, USA

**Keywords:** dairy cows, stressors, immune modulation, rich grain TMR, SARA

## Abstract

**Simple Summary:**

Dairy cows are often exposed to stressors during the lactation cycle. Nutritional stressors could be caused by rich-grain diet, leading to ruminal pH reduction and subsequent systemic inflammation. This metabolic pathology impacts animal health and productivity. Feed additives could provide beneficial effects on innate immune function in dairy cows, especially during stressing periods. The goal of this study was to determine the effect of OmniGen-AF on measures of immunity, inflammation, and liver function in lactating dairy cows fed a high-starch, low-fiber diet. Changes in rumination, pH, and volatile fatty acids were recorded. Treated cows resulted in better rumen volatile fatty acids profile and also showed shifts in hematological parameters compatible with a prompter regeneration of red blood cells, greater proportion of neutrophils, lower levels on GGT, PON, and BHB. These results show evidence of the nutritional stress induced by feeding a high-starch, low-fiber diet, and suggest that the fed additive tested modulates some of the metabolic and immunological responses to sub-acute ruminal acidosis.

**Abstract:**

Dairy cows are often exposed to multiple stressors in a lactation-cycle, with sub-acute ruminal acidosis (SARA) a frequent example of nutritional stress. SARA affects ruminal and intestinal equilibrium resulting in dysbiosis with localized and systemic inflammation impacting animal health and productivity. OmniGen-AF (OMN, Phibro Animal Health Corporation, Teaneck, NJ, USA) is a feed product recognized for modulating innate immune function, especially during periods of stress. The objective of this study was to determine the effects of OMN in lactating dairy cows fed a high-starch, low-fiber diet. Twenty-four blocked cows were assigned to control or treatment (55 g/d). After the additive adaptation (49 d) cows were fed the challenge diet (28 d). Milk, rumination and pH were continuously recorded; components, rumen fluid, and blood were taken in multiple time-point and analyzed. Results showed that the challenge decreased the rumination, shifted ruminal fluid composition, decreased milk production and the components, and slightly increased the time below pH 5.5, with no differences between groups. The treatment produced greater rumen butyrate and lower lactate, prompter regeneration of red blood cells, increase of neutrophils, lower paraoxonase, gamma-glutamyl-transferase, and β-hydroxybutyrate, with no differences on other tested inflammatory markers. Results show that OMN helps modulating some of the metabolic and immunological responses to SARA.

## 1. Introduction

Dairy cows are exposed to multiple stressors during their life [1], especially during the periparturient period [2,3,4] and lactation [5]. Examples of stressors experienced by the high producing dairy cow include overcrowding and group changes, high environmental temperature, and feeding errors. The effect of these stressors negatively influences productive and reproductive performance [1]. Moreover, stress may affect metabolism and immune functions [6,7]. Severe or chronic stress disrupts homeostasis, altering biological functions and predisposing animals to several pathologies [8].

Digestive disorders created by high-grain diets and lack of physically effective fiber from forages are the main responsible for the subacute ruminal acidosis syndrome (SARA) [9,10,11], that negatively affects performance resulting in substantial economic losses to farmers [12].

An impaired rumen epithelium is not able to prevent the simultaneous entry of microbes and luminal toxins into the systemic circulation [13,14]. SARA could lead to a failure in the selective rumen epithelium barrier function, thereby enabling luminal immunogens to translocate into the blood supply and lymphatic system [15]. More specifically, various luminal toxins such as endotoxins and biogenic amines seem to interfere with the epithelial constraint function by altering the structure and function of the tight junction barrier, thereby disrupting the integrity of epithelial cells and enabling their translocation changing cellular pathways [14]. There is a growing body of evidence that indicates that SARA leads to enhanced growth and lysis of Gram-negative bacteria followed by the release of great amounts of lipopolysaccharides (LPS) [16,17]. The gastrointestinal LPS, is a potentially pro-inflammatory molecule that has been investigated widely relatively to the immune system [16,18]. Once gastrointestinal LPS enters into circulation, a pro-inflammatory cascade is triggered and characterized by moderate elevation of serum acute-phase proteins (APP) with LPS-binding protein, interleukins, and serum amyloid A [19] being the main markers of the ruminal LPS translocation in cattle [17,20].

The development of uncontrolled acute or chronic inflammatory responses to LPS may not only cause damage to host tissues but also costs in terms of energy, changing the prioritizations of nutrients, influencing the energy balance, and reducing animal growth and productivity (1). The first barrier against pathogens is represented by gastro-intestinal epithelia (14), secondary the immune system [21], which is fundamental in the modulation of the inflammatory response. The resilience provided by an efficient immune system is essential to limit disorders and consequently, the use of antimicrobial drugs in food producing animals [22], in line with the guidelines suggested by EU legislation [23].

Different nutrients are known to modulate the response of the immune system, and commercial preparations are available, e.g., OmniGen-AF (OMN; Phibro Animal Health Corporation, Teaneck, NJ, USA). This blend of ingredients is a combination of several ingredients of which yeast cell wall material is one. The yeast cell wall does contain mannans and glucans which are known to stimulate or activate an immune response through the gastro-intestinal tract, and they have been shown to benefit rumen fermentation. The use of yeast cell wall also has been shown to reduce inflammation during pathogenic, physiological, and immunological challenges [24,25]. In actual fact, OMN has been effective in supporting immune function in dairy cattle under different stressors, even if the intimate mechanisms of action have not been fully elucidated. Brandão et al. [26] administered LPS to lactating cows and found greater levels of haptoglobin and tumor necrosis factor alpha when animals were fed OMN. Ortiz-Marty et al. [27] reported maintained neutrophil function in OMN fed mice during a dexamethasone and LPS challenge. Dry cows fed OMN during heat stress had greater mammary gland tissue regeneration and produced more milk [28]. Mezzetti et al. [29] observed OMN promoted rumination recovery, and reduced lipid mobilization and ketogenesis. Mammi et al. [30] evaluated the effect of OMN supplementation in dairy cows and found fewer health events, lower somatic cell count (SCC), and reduced involuntary culling rate in treated animals.

Despite the evidence of OMN supplementation effects listed in previous studies [26,27,28,29], to our knowledge, no studies have analyzed the effects of this immunomodulant product in lactating dairy cows exposed to a high concentrate diet challenge in a longitudinal trial. The aim of our study was therefore to study the mitigation effects in zootechnical and immunological parameters.

## 2. Materials and Methods

### 2.1. Experimental Design, Housing and Diets

This study was conducted at the University of Bologna dairy research farm. The experimental design is outlined in Figure 1.

According to the capability of the University of Bologna dairy and research barn, twenty-four lactating multiparous Italian Holstein-Friesian cows were distributed in two treatment groups balanced by parity, DIM, milk yield and components (Table 1), and the groups were randomly assigned to treatment. Cows in the OmniGen AF treatment (OMN) were fed 55 g/d of OmniGen AF (Phibro Animal Health Corporation, Teaneck, NJ, USA) and cows in the control treatment (CON) received no supplement.

The experiment consisted of three phases listed as covariate, pre-challenge, and challenge phase. In the pre-trial and pre-challenge phase cows were housed in free-stall pens (OMN or CON) and group fed a TMR (Table 2); during the covariate phase, cows were sampled and data recorded in order to balance groups. After that, OMN was supplied since the beginning of the trial (pre and challenge periods) and mixed into the TMR of the OMN group. The ration was formulated to mimic a standard Parmigiano Reggiano ration, based on dry forages and approved concentrates, and it was balanced using a software based on the CNCPS model (DinaMilk5; Fabermatica, Ostiano, Italy).

Diets were mixed and fed once daily at 0900 h and offered ad libitum (approx. *1.1 expected intake). Additionally, grass hay was available ad libitum during the covariate and pre-challenge phase and not available during the SARA challenge phase. Cows were milked twice daily in a 2 × 5 herringbone parlor. Milk yield and BW were recorded at every milking (Kibbutz Afikim, Israel). The covariate phase lasted 14 d. The pre-challenge phase lasted for 49 d, which is the time that previous research [33] has shown is needed to demonstrate differences in immune function to feeding OMN; in Wu et al. [15], this time was effective in increasing the gene neutrophil expression of the adhesion molecule SELL and the cytokine CXCL8. The pre-challenge phase was followed by a 28 d SARA challenge phase. Cows were exposed to the challenge in three consecutive time blocks of eight cows each (four cows per treatment). During the challenge, cows were housed in tie-stalls bedded with sawdust, and they had free access to individual feed bunks and water dispensers. The diet consisted of a TMR made with the same ingredients fed during the pre-challenge phase, but the inclusion rates of some ingredients were modified to increase starch, while decreasing aNDFom, peNDF, and uNDF240 (Table 2). The starch raised from 22.95 to 33.62% DM thanks to the increase in corn flakes content (from 6 to 13 kg/cow/day as fed). Fibrous fractions decreased from 35.94, 17.56, and 9.93 to 29.00, 13.80, and 8.05% DM of aNDFom, peNDF, and uNDF240, respectively, thanks to the reduction of grass hay (from 9.5 to 6 kg/cow/day as fed). OMN (55 g/d) was top dressed to the corresponding cows immediately after the ration delivery. DMI was measured by weighing feed offered and orts, and water intake was automatically recorded by flow meters. For milking, cows were moved to the same milking parlor previously described; each milking lasted for approx. 45 min.

### 2.2. Feed and Milk Sampling

Samples of feedstuffs, diets, and orts were collected twice weekly throughout the experiment (Mondays and Thursdays), dried in a forced-air oven at 65 °C. Samples were firstly checked by NIR techniques (TANGO FT-NIR Spectrometer, Bruker Optics GmbH, Ettlingen, Germany, [34]) and analyzed for DM, CP, aNDFom, ADF, peNDF, and starch as previously described [35,36]. In vitro aNDFom digestibility (24 h and 240 h) was determined in buffered media containing ruminal fluid [37]. Digestibility was performed on forages and TMR according to the procedure described by Palmonari et al. [38]. In vitro aNDFom digestibility at 240 h was performed using the Tilley and Terry modified technique [39]. Milk samples from two consecutive milkings from each cow were collected on d −14 and −3 prior to start of the experiment, on d 0, 7, 14, 21, and 28 of the SARA challenge (Figure 1) and analyzed by a certified laboratory (Associazione Provinciale Allevatori Bologna) for fat, total protein, lactose, and SCC. ECM was then calculated.

### 2.3. Rumen Sampling and Measurements

Cows were monitored for reticular pH with an indwelling wireless transmitting unit (SmaXtec Animal Care Sales GmbH, Graz, Austria), a system previously validated in rumen-cannulated dairy cows [40]. These devices (3.5 cm i.d., 12 cm long, and weighing 210 g) were calibrated following the manufacturer instructions and manually inserted into the rumen via the esophagus one week before the start of the pre-challenge period. Previous research has showed that these devices tend to sit in the ventral reticulum area [40]. pH and temperature were recorded every 10 min and data transmitted real-time to a base station using the ISM band (433 MHz). Data were then collected using an analog-to-digital converter and stored in an external memory chip. Reticular pH data were aggregated as daily means, and a pH threshold of 5.5 was used to calculate time and dispersion below that threshold [41,42]. Rumen fluid was collected via esophageal tube at 0845 h on d 0, 14, and 28 of the SARA challenge. The first 500 mL of rumen fluid collected were discarded before taking samples. Rumen fluid was analyzed for volatile fatty acid (VFA) concentrations by gas chromatography [43]; ammonia was assessed using a commercial kit (urea/BUN—color, BioSystems S.A. Barcelona, Spain); and l-lactic acid and d-lactic acids were determined with a commercial kit (K-DLATE, Megazyme Co., Wicklow, Ireland). Commercial standards were used for the calibration of the kits. Rumination time was continuously monitored during the entire experiment using the Hi-Tag rumination monitoring system (SCR Engineers Ltd., Netanya, Israel).

### 2.4. Blood Sampling

Blood was collected from the coccygeal vein at 0845 h on d −14, −7, and −3 prior the start of the experiment, on d 0, 1, 2, 3, 7, 14, 21, and 28 of the SARA challenge. Samples were taken into vacuum tubes containing either EDTA (for complete blood counts), clot-activator (silicate, for serum assays), or Li-heparin (for plasma assays) (Vacutest, Kima, Padova, Italy). EDTA tubes were kept at 4 °C after collection and blood counts were performed within 4 h. Clot-activator and Li-heparin tubes were centrifuged at 2000× *g* for 20 min and 3000× *g* for 10 min to obtain serum and plasma, respectively (Centrifugette 4203, ALC International Srl, Cologno Monzese, Italy). Serum and plasma samples were stored at −80 °C until analysis. Complete blood counts (CBC) were performed at the Clinical Pathology Laboratory University of Bologna Veterinary Hospital using an automated hematology system (ADVIA 2120, Siemens Healthcare Diagnostics, Tarrytown, NY, USA) according to previous studies [44,45]. The CBC listed several parameters: hemoglobin (HG), haematocrit (HTC), erythrocytes (ERT), reticulocytes (RET), mean corpuscular haemoglobin concentration (MCHC), mean corpuscular volume (MCV), red cell distribution width (RWI), leukocytes (LEU), lymphocytes (LYM), neutrophils (NEU), and eosinophils (EOS). Plasma samples were analyzed at the Università Cattolica del Sacro Cuore (Piacenza, Italy): a clinical auto-analyzer (ILAB-650, Instrumentation Laboratory, Lexington, MA) was used to determine the concentration of beta hydroxybutyrate (BHB), gamma-glutamyl transferase (GGT), haptoglobin (HAPT), ceruloplasmin (CRP), albumin (ALB), and cholesterol (CHOL) following Calamari et al. [46]. Reactive oxygen metabolites (ROM) and ferric reducing antioxidant power (FRAP) were determined according to Jacometo et al. [47]; and paraoxonase (PON) was determined according to Bionaz et al. [48]. Calibrations were performed through commercial standards for CRP, ALB, BHB, ROM and FRAP, and through internal standards for the rest. Four different quality controls were used to test the repeatability and precision for each parameter. Furthermore, plasma samples were used to determine IL-1ß and serum amyloid A (SAA) using a multi-detection microplate reader (BioTek Synergy 2, Winooski, VT, USA) and commercial ELISA kits specific for the bovine species (Pierce, Thermo Scientific, Rockford, IL, USA) for IL-1ß, or TP-802 (Tridelta D.L., Ireland) for SAA. Serum samples were analyzed at the Istituto Zooprofilattico Sperimentale della Lombardia e dell’Emilia Romagna (Brescia, Italy). Commercial kits were used on these samples to measure IL-6 (DuoSet ELISA, cat. no. DY8190, R & D Systems, Minneapolis, MN, USA) and γIFN (BovigamTM TB Kit, cat. no. 63,320, Thermo Scientific Prionics AG, Schlieren, Zurich, Switzerland). In both cases, the calibrations were performed through standard solutions according to the manufacturer’s instructions. Full methodologies including the coefficient of variation are reported in Calamari et al. [46].

### 2.5. Data Analysis

Data were analyzed using the software JMP v15.1 (SAS Institute Inc., Cary, NC, USA). Linear mixed effects models were used. Model main fixed effects were treatment, diet and interaction. Data related to reticular pH and temperature, rumination time, milk yield, and BW were analyzed considering only the last 28 d of the pre-challenge phase in order to balance the model. Data related to rumen fluid parameters, milk components, and blood parameters were analyzed as repeated measurement (first-order autoregressive AR1) considering the sampling day as the time effect, and using the baseline established prior to the start of the experiment (d −14, −7, and −3) as covariate depending on scheduled sampled (Figure 1). Data of DMI and WI were recorded only during the tie stall period; thus, only the treatment effect was considered. A preliminary analysis including blocks as fixed effect was conducted and resulted in no significance; thus, this factor was included as nested effect into the random factors. Each cow within block and treatment was considered as experimental unit and used as random variable for all analyses. Normal distribution of the data was checked for the residuals resulted from an initial mixed model, and normalized, when necessary, by BoxCox transformation. Means are reported as least square mean and pairwise multiple comparisons were performed using Student *t*-test as post hoc test when a *p*-value ≤ 0.10 was detected. A *p*-value ≤ 0.10 was considered a tendency; a *p*-value ≤ 0.05 was considered statistically significant; and a *p*-value ≤ 0.01 was considered highly significant.

## 3. Results

### 3.1. Intake and Milk Production

As formulated, the diet fed during the SARA challenge contained, vs. the standard diet, considerably more starch (34% vs. 23%), and less aNDFom (29% vs. 36%), peNDF (14% vs. 18%), and uNDF240 (8.1% vs. 9.9%) (Table 2). The results of intakes, BW and production are reported in Table 3. No differences were detected between the treatment groups during the SARA challenge neither on DMI (25.8 vs. 25.7 kg/d in CON and OMN, respectively, *p* = 0.99) nor water intake (144 vs. 147 L/d in CON and OMN, respectively, *p* = 0.70). Milk yield was similar between treatment groups (41.7 and 42.0 kg/d for CON and OMN, respectively, *p* = 0.94) but decreased equally in both groups in the SARA challenge period (−2 kg/d of milk, *p* < 0.01). A drop in milk fat and total protein yields were recorded during the SARA challenge (−287 and −93.4 g of fat and total protein yields, respectively, *p* < 0.01) with no differences between treatments (1310 vs. 1223 and 1332 vs. 1367 g of fat and total protein yields in CON and OMN, respectively, *p* = 0.35 and 0.73). Somatic cell count and MUN did not vary between treatments but the somatic cell count increased (+0.59_log10 cells × 1000/mL_, *p* < 0.01) and MUN decreased (−3.9 mg/dL, *p* < 0.01) when the SARA challenge diet was fed. BW did not change throughout the experiment or between treatment groups (644 and 628 kg, in CON and OMN, respectively, *p* = 0.61).

### 3.2. Ruminal Parameters

Results of rumination time and reticular pH and temperature are reported in Table 3. Rumination time decreased during the SARA challenge (−76 min/d, *p* < 0.01) in both treatment groups with no differences between treatments. No significant differences were observed in daily average reticular pH neither between diets nor treatment groups. However, time below pH 5.5 increased in both treatment groups going from the pre- to the challenge diet (+19.26 min/d, *p* < 0.01). Even if very slightly, reticular temperature increased during the SARA challenge diet (+0.01 °C, *p* < 0.01). Results of VFA concentrations are reported in Table 4. Total rumen VFA concentrations and propionic acid increased when fed the challenge diet (+16.2 mmol/L of VFAs, *p* < 0.01 and +5.6% mmol of propionic acid, *p* < 0.01), while acetic and nor-butyric acid decreased (−3.3% mmol of acetate, *p* = 0.03 and −1.9% mmol of nor-butyrate, *p* < 0.01). The treatment did not modify total VFA (101.0 vs. 99.3 mmol/L in CON and OMN, respectively, *p* = 0.71) and ac. Acetic (53.2 vs. 55.0% mmol in CON and OMN, respectively, *p* = 0.13). The use of OMN was associated with the decrease of propionic acid (*p* = 0.04) mainly on d 14 (−6.3% mmol, *p* ≤ 0.10), and increased iso-butyrate proportion on d14 (+0.11% mmol, *p* ≤ 0.05). Additionally, both l- and d-lactic acid were lower in OMN animals since the beginning of the SARA challenge period (−32 mg/dL and −65 mg/dL, *p* = 0.09 and *p* = 0.02, respectively).

### 3.3. Hematological, Metabolic, and Immunological Parameters

The SARA challenge had strong effects on metabolic, health, and immunological parameters in all cows (Figure 2, Figure 3, Figure 4, Figure 5 and Figure 6). Most of RBC, WBC, inflammatory, metabolic, and oxidative status parameters were affected by the SARA challenge. In particular, the levels of HG, HTC, ERT, MCHC, LEU, NEU, CHOL, ALB, BHB, and FRAP declined progressively along the SARA challenge period, while MCV, RWI, EOS, CER, SAA, ILs, GGT, PON, and ROM increased. Among CBC parameters, OMN increased RET (0.061 vs. 0.045%, in OMN and CON, respectively, *p* = 0.09, Figure 2d) and NEU (44.88 vs. 40.26%, in OMN and CON, *p* = 0.02, Figure 3b), while LYM resulted decreased (46.32 vs. 50.58%, in OMN and CON, *p* = 0.02, Figure 3c). A treatment effect was also shown on inflammatory and metabolic parameters: CORT increased (11,537 vs. 9319 pg/mL, in OMN and CON, respectively, *p* = 0.02, Figure 4a), while PON (96.75 vs. 87.65 U/mL, in CON and OMN, respectively, *p* = 0.01, Figure 6a) and GGT (−2.16, U/L, in OMN, *p* = 0.01, Figure 5d) resulted lower in OMN cows compared to CON cows. Finally, BHB, compared to CON cows, was lower in OMN cows during the last days of the challenge (0.49 vs. 0.40 mmol/L d21 and 0.48 vs. 0.41 mmol/L d 28, in CON and OMN, respectively, *p* = 0.05, Figure 5c).

## 4. Discussion

The present trial deals with the mitigation of the stress responses to a high concentrate diet challenge in lactating dairy cows by the supplementation of an immunomodulatory feed additive.

Regarding the results showing the effect of the challenge in the enrolled animals the rumination time, rumen pH, production, and blood markers changed consistently. The decrease in rumination time in response to the dietary challenge may be related to the levels of starch and aNDFom in the diet (34% and 29% of DM, respectively), rather than the content of peNDF (13.8% of DM). The latter was lower than min. levels recommended by other authors [49], but previous experience with feeding diets similar to the one fed in this trial, a diet based on hay and straw as required in the Parmigiano Reggiano production area [50,51] has shown it is possible to decrease the level peNDF to 11.2% of DM without compromising rumen health [52,53,54]. In all those examples the starch content of the ration (avg. 23.2% DM) was lower than in the SARA challenge diet, and comparable to the pre-challenge diet, fed in this trial. At the same time, a min. safe level of 9% uNDF of DM has been recommended [54,55], while this was 8% in our SARA challenge diet. Therefore, a high starch content, combined with a low uNDF content, were likely the reasons for the drop observed in rumination time, which has been identified as a marker for SARA [56,57].

Daily mean reticular pH was not as low as expected and did not change because of the dietary challenge, but time below reticular pH 5.5 slightly increased (Table 3). Rumen pH thresholds suggested to indicate SARA were not reached in the present study (e.g., 330 min/d of pH < 5.6, [16]; or <5.8, [49]). However, the definition of rumen acidosis in terms of rumen pH thresholds is still under discussion and rumen pH cannot be seen as the sole marker for this digestive and metabolic disorder [11]. In addition, the absence of a marked decrease in pH during the challenge could also be related to the measurement system used: pH was recorded at the reticulum, and previous research has shown limited comparability between pH recorded at the reticulum and the rumen [58]. Mensching et al. [59] found a difference of about 0.4 points pH higher in the reticulum than in the rumen. Moreover, the pH in the reticulum is more stable compared to the pH in the rumen [60]. Other signs of SARA, all of which were seen in the present study, include milk yield decrease, milk fat depression, and inversion of fat-protein ratio [12,61]. All these reasons could be a limitation of our study.

Some immune and metabolic markers increased over the dietary challenge: EOS, CER, SAA, ILs, GGT, PON, and ROM. Acute phase proteins are produced mainly in the liver and are considered sensitive markers of inflammation. The positive APP, including CER and SAA, have a protective role against pathogens, e.g., in neutralizing enzymes, scavenging free hemoglobin and radicals, and in modulating the host’s immune response [62]. The increase in CER (Figure 4b), even if not specific, is an expression of a systemic and innate reaction of the organism to inflammation triggered by external (pathogens, toxins, etc.) or internal (tissue damage, etc.) stimuli [63]. Moreover, SAA (Figure 4c) is reported as a marker of the ruminal LPS translocation in cattle [17,20]. This increase could be related in our study to the dietary challenge, probably because of a translocation of LPS out of the digestive tract into the portal circulation [17,63]. The increase in positive APP was also seen in previous nutritional challenge trials [64,65], in which an increase in plasma concentrations of positive APP were observed when rumen pH was below 5.8 for at least 6 h a day. On the contrary, ALB, a negative APP, slightly decreased over the dietary challenge (Figure 5a) which is likely the consequence of a shift towards production of positive APP in the liver [66,67]. CHOL, another marker of cow wellbeing, diminished in the challenge phase of our study (Figure 5b) and this supports the impact of the dietary challenge on immune function and metabolism. Previous studies [68] reported a lower level of plasma CHOL in farms with high prevalence of SARA; CHOL was used in that study as an index of the CHOL binding protein, a negative APP related to an inflammatory response.

In addition, the challenge diet was associated with a decrease in red blood cell (RBC) parameters, mostly HG, HTC, ERT, MCV, MCHC, and RWI (Figure 2), highlighting the stress experienced by the animals. During chronic stress mature red blood cell forms decrease and more immature forms (RET) are released [69]. RET usually have higher volume and lower content in HG, which are likely the reasons for the decrease observed in MCHC and the increase in RWI in this trial. Altogether, the changes observed in rumination parameters and production, and in metabolic and immune markers suggest that the diet fed during the SARA challenge was effective in creating the intended nutritional, metabolic and immunological stress.

Regarding the effect of the immunomodulant product in treated animals limited interesting results has been collected. Cows in the OMN treatment tended to have a higher percentage of RET throughout the nutritional challenge (Figure 2d), suggesting a prompter response in in replacing damaged ERT with immature red blood cells (RET). Mezzetti et al. [29] observed similar effects on RBC in transition cows fed OMN. As reported by other authors, OMN exerts effects on white blood cells [15,24,33,70]. In the present study, we observed a change in the proportion of white blood cells species in OMN treated cows, with a decrease in LYM (Figure 3c) compensated by an increase in NEU (Figure 3b). These results are consistent with previously reported findings related to an increase in NEU and phagocytic activity of polymorphonuclear cells upon feeding OMN [15,71,72]. Other studies have reported positive effects of this product on leukocyte function and gene expression of L-selectin [24,73,74]. CORT, another stress marker, was greater in the OMN treatment, another finding that under certain circumstances has been previously reported for this product by other authors [24,71,72] and suggests some modulation of the innate immune system [75,76]. On the other hand, on some previous studies the supplementation of OMN was associated by equal or lower levels of CORT [29,33,74]. These differences on CORT recorded levels could be related with the high variability of this parameter and low sampling frequency applied in this research. Moreover, it is proposed that the functional metabolites, organic acids, vitamins, and antioxidants present in the yeast cells’ wall, one of the active ingredients in OMNG, may either be used as nutrients by the rumen and gut microbiota or act as signaling molecules affecting interactions between microbes and the immunological response [77].

During the SARA challenge BHB gradually decreased in all animals, probably because of the highly energetic diet. However, this reduction was more evident in OMN fed cows during the second half of the challenge, becoming significantly greater than in CON cows at d 21 and 28 (Figure 5c). This suggests a better energy balance of OMN fed cows, as previously suggested by Wu et al. [15,78] and could be explained by the energetic cost of inflammation. Inflammation changes the prioritization of nutrients, affecting energy balance and performance [22]. At the same time, depletion of energy storages, excess of NEFA and BHB result in fatty liver and ketosis which have negative effects on the immune function [21,79] and productive performance [80] of early lactating cows. GGT, a marker related with the liver function and index of cholestasis [81] was lower in OMN fed cows during the challenge, suggesting healthier liver function in those cows (Figure 5d).

As antioxidant capacity directly depends on liver activity [22], dysregulation on the liver functions reflects on blood concentrations of such biomarkers. Among oxidative status markers, PON, which is also a negative APP, was lower in the OMN treatment (Figure 6a) although levels observed were higher than the minimum indicated by Trevisi and Minuti [62], suggesting no depletion of this compound. ROM had no overall variation due to the treatment (Figure 6c); different results were reported by Mezzetti et al. [29] where this marker was found at lower levels in OMN fed cows. Finally, FRAP was not affected by the treatment (Figure 6b); this compound is known to exert antioxidant activity and provides a measurement of antioxidant power via blood concentration of bilirubin, uric acid, proteins, and vitamins C and E [82].

Finally, in literature the effects of the tested immunomodulatory feed additive are greater during the transition period [15,28,29], a really risky phase for dairy cows [4]. In the present research, the enrolled cows were, at the beginning of the challenge period, around 100 DIM. In this phase their physiological status is more stable and less susceptible to external stressors [1].

## 5. Conclusions

In conclusion, these results show evidence of the nutritional stress induced by feeding a high-starch, low-NDF challenge diet, with measurements of digestive, metabolic, and immunological markers. The digestive impact was markedly seen as a decrease in rumination time and a shift in acetate and propionate proportions, even if reticular pH was barely impacted. Not all metabolic and immunological markers were impacted to the same degree but CER, GGT and ROM increased while ALB and BHB decreased along the challenge, reflecting the metabolic and immunological impact of this type of diet. Cows fed OMN showed a modulated metabolic and immune response to the challenge diet, as reflected by hematological changes compatible with a more reactive regeneration of red blood cells, a greater proportion of neutrophils in WBC, higher CORT, and lower PON, GGT, and BHB.

## Figures and Tables

**Figure 1 animals-12-02129-f001:**
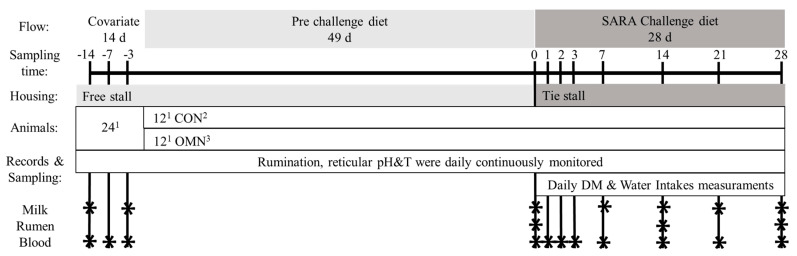
Experimental design. ^1^ Divided on 3 consecutive blocks. ^2^ Control group. ^3^ Treated group (cows receiving OmniGen-AF, 55 g/d).

**Figure 2 animals-12-02129-f002:**
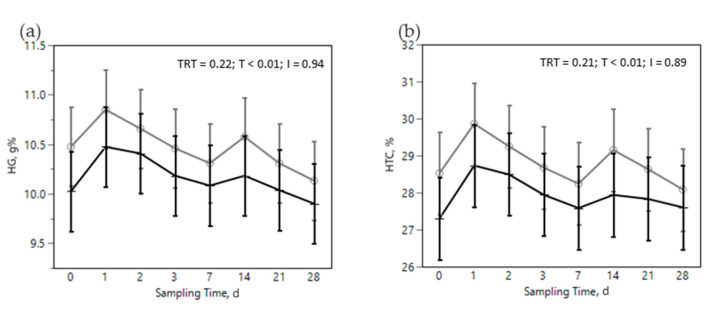
Effect of the dietary challenge on red blood cells parameters: hemoglobin (HG, g%, (**a**)), hematocrit (HTC, %, (**b**)), erythrocytes (ERT, n°/µm (**c**)), reticulocytes (RET, %, (**d**)), mean corpuscular volume (MCV, fL, (**e**)), mean corpuscular hemoglobin (MCHC, g%, (**f**)) and red cell distribution width (RWI, %, (**g**)) in cows in the CON ^1^ (
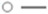
) or OMN ^2^ (
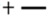
) treatments during the SARA challenge. ^1^ Control group. ^2^ Treated group (cows receiving OmniGen-AF, 55g/d). TRT: treatment *p-*value effect; T: time *p-*value effect; I: interaction TRT x T *p-*value effect.

**Figure 3 animals-12-02129-f003:**
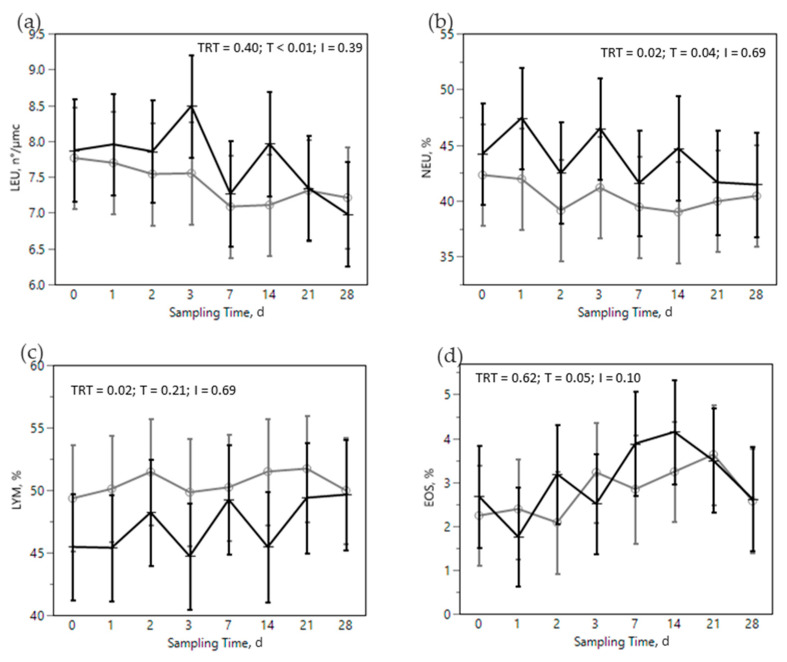
Effect of the dietary challenge on white blood cells parameters: leukocytes (LEU, n°/µmc, (**a**)) neutrophils (NEU, %, (**b**)), lymphocytes (LYM, %, (**c**)) and eosinophils (EOS, %, (**d**)) in cows in the CON ^1^ (
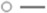
) or OMN ^2^ (
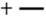
) treatments during the SARA challenge. ^1^ Control group. ^2^ Treated group (cows receiving OmniGen-AF, 55g/d). TRT: treatment *p-*value effect; T: time *p-*value effect; I: interaction TRT x T *p-*value effect.

**Figure 4 animals-12-02129-f004:**
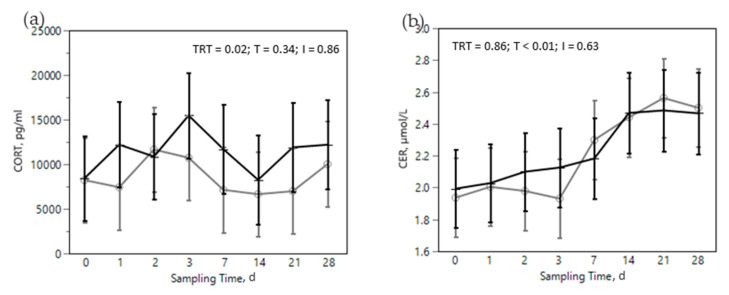
Effect of the dietary challenge on inflammatory markers: cortisol (CORT, pg/mL, (**a**)), ceruloplasmin (CER, µmol/L, (**b**)), Serum Amyloid A (SAA, µg/mL, (**c**)), Interleukin 1 beta (IL1β, pg/mL, (**d**)) and Interleukin 6 (IL6, pg/mL, (**e**)) in cows in the CON ^1^ (
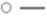
) or OMN ^2^ (
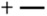
) treatments during the SARA challenge. ^1^ Control group. ^2^ Treated group (cows receiving OmniGen-AF, 55g/d). TRT: treatment *p-*value effect; T: time *p-*value effect; I: interaction TRT x T *p-*value effect.

**Figure 5 animals-12-02129-f005:**
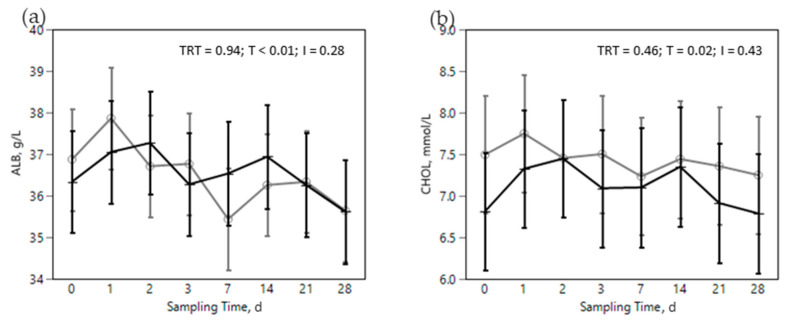
Effect of the dietary challenge on metabolic status markers: albumins (ALB, g/L, (**a**)), cholesterol (CHOL, mmol/L, (**b**)), beta hydroxybutyrate (BHB, mmol//L, (**c**)) and gamma glutamyl transferase (GGT, U/L, (**d**)) in cows in the CON ^1^ (
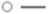
) or OMN ^2^ (
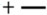
) treatments during the SARA challenge. ^1^ Control group. ^2^ Treated group (cows receiving OmniGen-AF, 55g/d). TRT: treatment *p-*value effect; T: time *p-*value effect; I: interaction TRT x T *p-*value effect. ** *p*-value ≤ 0.05 between TRT within time point.

**Figure 6 animals-12-02129-f006:**
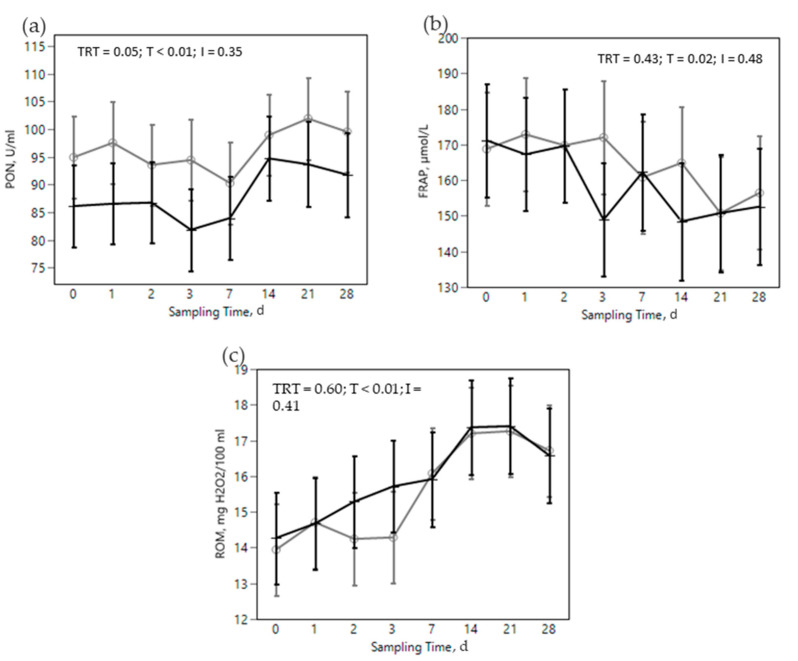
Effect of the dietary challenge on oxidative status markers: paraoxonase (PON, U/mL, (**a**)), Ferric reducing antioxidant power (FRAP, µmol/L, (**b**)), and reactive oxygen metabolites (ROM, H2O2/100mL, (**c**)) in cows in the CON ^1^ (
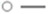
) or OMN ^2^ (
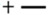
) treatments during the SARA challenge. ^1^ Control group. ^2^ Treated group (cows receiving OmniGen-AF, 55g/d). TRT: treatment *p-*value effect; T: time *p-*value effect; I: interaction TRT x T *p-*value effect.

**Table 1 animals-12-02129-t001:** Characteristics of the cows assigned to CON and OMN treatment groups at the beginning of the experiment (Covariate) and before the challenge period (T0) (mean ± SD).

Cows’ Characteristics	Beginning Experiment		Beginning Challenge	
	CON	OMN ^1^	CON	OMN ^1^
Age, y	2.65 ± 0.66	2.62 ± 0.52	2.78 ± 0.79	2.75 ± 0.65
Lactation, n	1.64 ± 0.65	1.67 ± 0.65	1.64 ± 0.65	1.67 ± 0.65
DIM ^2^	51.5 ± 28.9	52.3 ± 30.5	100.5 ± 28.9	101.3 ± 30.5
BW ^3^, kg	630 ± 58.3	633 ± 64.1	644 ± 73.2	634 ± 70.4
Milk yield, kg/d	40.0 ± 7.74	40.4 ± 7.78	41.77 ± 6.85	43.0 ± 9.06
Fat, %	3.91 ± 0.74	3.89 ± 0.54	3.59 ± 0.18	3.53 ± 0.15
Total protein, %	3.30 ± 0.24	3.28 ± 0.23	3.24 ± 0.18	3.26 ± 0.20
Lactose, %	4.95 ± 0.17	5.02 ± 0.10	4.96 ± 0.17	5.03 ± 0.08
MUN ^4^, mg/dL	8.27 ± 2.90	8.35 ± 3.26	9.79 ± 1.99	9.15 ± 1.99
SCC ^5^, log_10_/mL	1.45 ± 0.32	1.63 ± 0.42	1.78 ± 0.29	2.20 ± 0.32

^1^ OmniGen-AF supplemented at 55 g/d. ^2^ Days in milk. ^3^ Body weight. ^4^ Milk urea nitrogen. ^5^ Somatic cell count.

**Table 2 animals-12-02129-t002:** Composition of experimental diets.

Diets’ Composition	Pre-Challenge Diet	SARA Challenge Diet
Ingredients ^1^, kg/cow/d, as fed		
Grass hay, finely chopped	9.5	6.0
Wheat straw, finely chopped	1.0	1.0
Corn flakes	6.0	13.0
Concentrate ^2^	7.5	8.0
Liquid feed ^3^	1.0	1.0
Grass hay, long	Ad libitum	-
Forage:Concentrate	45.4:54.6	24.8:75.2
Chemical composition, %DM		
DM	87.22 ± 3.00	88.11 ± 0.74
Ash	7.50 ± 1.28	6.25 ± 0.43
Ether extract	3.21 ± 0.47	2.78 ± 0.65
aNDFom ^4^	35.94 ± 4.16	29.00 ± 2.80
ADF	24.55 ± 2.56	18.86 ± 1.64
ADL	5.27 ± 1.09	4.65 ± 0.19
uNDF_240h_ ^5^	9.93 ± 3.32	8.05 ± 0.96
Starch	22.95 ± 2.62	33.62 ± 2.45
peNDF_1.18mm_ ^6^	17.56 ± 1.35	13.80 ± 0.98

^1^ Additionally, 55 g/d of OmniGen-AF was added to the diet of cows in the OMN treatment. ^2^ Concentrate: 29.6% wheat bran, 29.4% sorghum grain, 21.6% canola meal, 14.7% flaked fullfat soybean, 2.2% calcium carbonate, 1% sodium chloride, 0.4% magnesium oxide, 0.9% sodium bentonite, and 0.3% vitamin and mineral premix (providing 40,000 IU/kg af vitamin A, 4000 IU/kg af vitamin D3, 30 mg/kg af vitamin E 92% α-tocopherol, 5 mg/kg af vitamin B1, 3 mg/kg af vitamin B2, 1.5 mg/kg af vitamin B6, 0.06 mg/kg af vitamin B12, 5 mg/kg af vitamin K, 5 mg/kg af vitamin H1 (para-aminobenzoic acid), 150 mg/kg af vitamin PP (niacin), 50 mg/kg af choline chloride, 100 mg/kg af Fe, 1 mg/kg af Co, 5 mg/kg af I, 120 mg/kg af Mn, 10 mg/kg af Cu, and 130 mg/kg af Zn). ^3^ Cane and beet pulp molasses blend fully characterized for composition, sugars and digestibility [31,32]. ^4^ Amylase- and sodium sulfite-treated NDF with ash correction. ^5^ Unavailable NDF estimated via 240 h in vitro fermentation. ^6^ Physically effective NDF (aNDFom*pef), calculated using the Ro-Tap system.

**Table 3 animals-12-02129-t003:** Effect of the dietary challenge on intakes, rumination time, reticular pH and temperature, milk yield and components in CON ^1^ or OMN ^2^ cows when fed the pre-challenge or SARA challenge diets.

Item	Pre-Challenge Diet		SARA Challenge Diet		SEM	*p*-Values		
	CON ^1^	OMN ^2^	CON	OMN		TRT	Diet	TRT x D
DMI, kg/d	-	-	25.8	25.7	1.25	0.99	-	-
WI, L/d	-	-	144	147	10.4	0.70	-	-
BW, kg	646	617	641	639	17.2	0.61	0.31	0.11
Rumination time, min/d	502	488	434	405	13.8	0.23	<0.01	0.27
Reticular pH	6.05	6.05	6.04	6.05	0.02	0.91	0.69	0.19
Reticular pH < 5.5, min/d	35.6	40.2	57.2	57.1	12.5	0.65	<0.01	0.64
Reticular temperature, °C	37.8	38.8	37.9	38.9	0.11	0.52	<0.01	0.91
Milk yield, kg/d	42.3	43.4	41.1	40.6	3.20	0.94	<0.01	0.12
Fat yield, g/d	1455	1365	1165	1081	9.48	0.35	<0.01	0.90
Protein yield ^3^, g/d	1351	1441	1312	1293	10.1	0.73	<0.01	0.11
ECM, kg/d	38.2	39.5	34.3	34.9	2.57	0.72	<0.01	0.35
MUN, mg/dL	10.13	9.16	6.04	5.46	0.72	0.23	<0.01	0.28
SCC, log_10_ cells/mL	1.78	2.20	2.45	2.71	0.29	0.52	<0.01	0.34

^1^ CON is control cows. ^2^ OMN is cows receiving OmniGen-AF (55g/d). ^3^ Total protein.

**Table 4 animals-12-02129-t004:** Evolution of rumen VFAs (acetic, propionic, iso-butyric, nor-butyric) and lactic acid concentration in CON ^1^ or OMN ^2^ cows when fed the pre-challenge or SARA challenge diets.

Item	D 0		D 14		D 28		SEM	*p-*Values		
	CON ^1^	OMN ^2^	CON	OMN	CON	OMN		TRT	Time	TRT x T
Total VFA, mmol/L	89.4	89.3	110.2	103.0	103.4	105.5	5.16	0.71	<0.01	0.62
Acetic, %mmol	56.6	56.0	52.5	55.5	50.5	53.5	1.34	0.13	0.03	0.30
Propionic, %mmol	27.8	28.4	36.0 ^a^	29.7 ^b^	37.1	32.0	1.76	0.04	<0.01	0.08
Iso-butyric, %mmol	0.50	0.36	0.36 ^B^	0.47 ^A^	0.42	0.47	0.06	0.12	0.69	<0.01
Nor-butyric, %mmol	13.1	12.3	9.8 ^b^	11.5 ^a^	10.6	11.2	0.55	0.33	<0.01	0.08
l-lactic, mg/dL	151	119	187	155	154	121	25.1	0.09	0.35	0.99
d-lactic, mg/dL	172	125	241	147	174	121	37.2	0.02	0.41	0.81

^1^ CON is control cows. ^2^ OMN is cows receiving OmniGen-AF (55g/d). ^A,B^ is *p* ≤ 0.05 and ^a,b^ is *p* ≤ 0.10 for differences among means within time point of sampling.

## Data Availability

The raw data supporting the conclusions of this article will be made available by the authors, without undue reservation.

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
