# Peer review of "Effect of an Immunomodulatory Feed Additive in Mitigating the Stress Responses in Lactating Dairy Cows to a High Concentrate Diet Challenge"

_animals, 2022, doi:10.3390/ani12162129_

Round 1

Reviewer 1 Report

General comments

1. Authors need to state what the objective of the study was in the introduction.

2. For all assays identified in the materials and methods please provide the inter and intra assay C.V.'s in order to assess the variability of the results.

Specific Comments

Page  Line                 Comment

6        245   It is redundant to state "no significant differences detected.  Just 

                   state "no differences were detected"

Author Response

Dear Reviewer 1,

thank you for your revisions.

  1. We stated the objective of the study in the introduction.
  2. We reported the reference including the inter and intra assay C.V.'s.
  3. We fixed the redundance on line 245

Reviewer 2 Report

The manuscript was very well written.

The immunomodulatory effects of OmniGen-AF were small. There are no effects on inflammatory or oxidative status markers. It should be extensively indicated in the simple summary, abstract, and the discussion

When the challenge started, DIM was around 100. However, there was no comment about it. Cow in the first weeks of lactation is exposed to multiple stressors, including diet changes. OmmiGen-AF effects maybe were not observed due to the reduced stress conditions that cows were exposed to. This could be indicated.

Author Response

Dear Reviewer 2, thank you for your revision. 

We changed as you suggested the simple summary, abstract, results and discussion.